# Wood Dust Flammability Analysis by Microscale Combustion Calorimetry

**DOI:** 10.3390/polym14010045

**Published:** 2021-12-23

**Authors:** Qiang Xu, Lin Jiang, Andrea Majlingova, Nikoleta Ulbrikova, Rhoda Afriyie Mensah, Oisik Das, Filippo Berto

**Affiliations:** 1School of Mechanical Engineering, Nanjing University of Science and Technology, Nanjing 210014, China; xuqiang@njust.edu.cn; 2Faculty of Wood Science and Technology, Technical University in Zvolen, 96053 Zvolen, Slovakia; majlingova@tuzvo.sk (A.M.); xszirmaiova@is.tuzvo.sk (N.U.); 3The Division of Material Science, Department of Engineering Sciences and Mathematics, Luleå University of Technology, 97187 Lulea, Sweden; rhoda.afriyie.mensah@ltu.se (R.A.M.); oisik.das@ltu.se (O.D.); 4Department of Mechanical and Industrial Engineering, Norwegian University of Science and Technology (NTNU), NO-7491 Trondheim, Norway

**Keywords:** wood dust, flammability, microscale combustion calorimeter, heat release rate

## Abstract

To study the practicability of a micro combustion calorimeter to analyze the calorimetry kinetics of wood, a micro combustion calorimeter with 13 heating rates from 0.1 to 5.5 K/s was used to perform the analysis of 10 kinds of common hardwood and softwood samples. As a microscale combustion measurement method, MCC (microscale combustion calorimetry) can be used to judge the flammability of materials. However, there are two methods for measuring MCC: Method A and Method B. However, there is no uniform standard for the application of combustible MCC methods. In this study, the two MCC standard measurement Methods A and B were employed to check their practicability. With Method A, the maximum specific heat release rate, heat release temperature, and specific heat release of the samples were obtained at different heating rates, while for Method B, the maximum specific combustion rate, combustion temperature and net calorific values of the samples were obtained at different heating rates. The ignition capacity and heat release capacity were then derived and evaluated for all the common hardwood and softwood samples. The results obtained by the two methods have significant differences in the shape of the specific heat release rate curves and the amplitude of the characteristic parameters, which lead to the differences of the derived parameters. A comparison of the specific heat release and the net calorific heat of combustion with the gross caloric values and heating values obtained by bomb calorimetry was also made. The results show that Method B has the potentiality to evaluate the amount of combustion heat release of materials.

## 1. Introduction

Wood dust as a side product of wood working (sawing, planing, milling and grinding) is a natural part of timber processing factories [1]. Wood dust poses a much higher risk than compact wood due to its flammable and explosive potentiality, for example, wood dust from grinding is extremely flammable and in certain conditions can create an explosive dust-air mixture. Once wood powder is ignited, it is easy to produce more gas and a lot of heat in a short time. At the same time, it will cause a larger area of wood dust to kick up dust, which will cause a large explosion. The combustibility of powder is the basis of dust combustion and explosion. In addition to the danger, wood dust has high energy and can be used as a high-quality fuel. The basic parameters, such as the combustion heat of wood dust, are helpful to support fire safety prevention and energy utilization [2,3].

In the field of micro scale calorimetry, most researchers used thermogravimetric analysis and differential scanning calorimetry to study the pyrolysis kinetics of combustible materials, which can be used as a reference to evaluate the flammability of materials [4,5,6]. In recent years, MCC has been developed and widely employed to evaluate the flammability of materials with advantages of only with milligram level samples and rapid heating rates. MCC is an experiment that integrates the concepts of pyrolysis, combustion and flow calorimetry to determine the composition and fire behavior of materials. The microscale combustion calorimeter was developed by the Federal Aviation Administration solely for screening aircraft materials and has existed for over a decade [6]. However, the equipment has become a key instrument for evaluating the flammability of all kinds of materials such as wood, textiles, plastics, etc. It is simple and wieldy with excellent repeatability and reproducibility as well as a long-life span. The range of sample sizes required is 0.5–50 mg with a sample heating rate of 0–10 K s^−1^ [7,8]. The MCC experiment uniquely produces separate burning stages, i.e., controlled pyrolysis of the solid material and combustion of the volatile gases. MCC adopts the principles of oxygen consumption calorimetry to measure the heat release rate, time and temperature of heat release. Based on the pyrolysis process applied during the experiment, these parameters can be calculated from the measurements; heat release capacity (HRC), net calorific value, specific heat release, pyrolysis and combustion residues, peak combustion and heat release temperatures [9,10,11,12,13,14,15,16,17]. It is worth mentioning that MCC cannot be used as a standalone fire test; a major limitation in its usage. MCC mostly assesses the heat release of a material and does not account for some fire characteristics such as melt, drip and smoke release [18].

According to the design and standards of MCC [7,8,9], there are two standard measurement methods, Method A [17] and Method B [18]. The Method A procedure, also referred to as controlled thermal decomposition, involves the pyrolysis of the sample in inert gas (nitrogen, helium, etc.) and complete oxidation of the volatile pyrolysis in a combustor. The rate and amount of heat release are obtained from the amount of oxygen consumed during the experiment. The HRC of combustion of volatile gases and pyrolysis residue are calculated from the results of Method A. Several studies have adopted this method to measure flammability parameters to define the fire behavior of various materials. On the other hand, the Method B process of the experiment is usually applied for combustion analysis. Combustion parameters such as the specific combustion rate, combustion residue and combustion temperature are measured. According to ASTM D7309-19 [19], controlled thermal oxidative decomposition is conducted by decomposing samples in a nitrogen and oxygen mixture prior to oxidation. Due to the nature of the pyrolysis and combustion processes, Method A allows for char formation while Method B ensures the complete combustion of residue. The similarity in the two approaches is that the pyrolysis gas will both burn up in air atmosphere after they escape from the pyrolysis room. However, according to the literature so far, almost all users use Method A to evaluate the flammability of the prepared materials, and there is no literature comparing Method A with Method B. However, Method B is more reasonable in the oxidizing atmosphere of combustible combustion since all the procedures of Method B are exposed to the oxidizing atmosphere.

A comparison of results measured from Method A and Method B has been made by researchers to determine the existing differences and similarities. Zhuge et al. [17] attempted the use of MCC as a tool to measure flame retardant mechanism efficiency in materials. In their work, they compared data repeatability for both Method A and B. According to the study, peak heat release rate (PHRR) repeatability for PC/diphosphate system was better in Method B than A whereas Method A showed more sensitivity in a PC/sulfonate salt system. They therefore concluded that data repeatability is material dependent. Similarly, the repeatability of PHRR and HRC from MCC were evaluated for nine different textile fabrics by Yang et al. [20]. This was done by comparing the standard deviation (SD) and coefficient of variance (CV) of the heat release properties. It was observed from the results that no significant differences were obtained for the SD and CV of PHRR and HRC from the two methods. However, it was stated emphatically that Method B results in a faster decomposition and heat release. A comparison of the parameters measured from Method A and Method B tests of three layers of cross laminated timber and two types of façade materials was made by Solorzano et al. [21]. The parameters considered in their research were HRC, total heat released (THR), char yield, heat of combustion (HOC), temperature at peak heat release (p_Temp_) and p_HRR_. From the experimental study of the façade materials, higher HRC and PHRR values were recorded for Method B while the values for the remaining parameters were similar in both pyrolysis modes. However, for the cross laminated timber, lower HRC, p_HRR_, THR, and HOC were measured in Method A, p_Temp_ was the same in all atmospheres and higher char yield results were recorded during the Method A test.

It is necessary to explore method B and compare the similarities and differences between the two methods, further improve the MCC experimental methodology, and promote the understanding of material flammability. In this paper, the basic flammability parameters of 10 kinds of Slovakian hardwood and softwood dusts were measured by using two MCC standard methods, especially the difference of the experimental results obtained by the two MCC measurement methods was compared. This paper has a good evaluation on the MCC calorimetric evaluation method and has a high-guiding significance in wood flammability.

## 2. Experiment Arrangement

### 2.1. Test Method

MCC tests were conducted with MCC-3 (manufactured by Deatak Inc., McHenry Illinois, IL, USA) located at Nanjing University of Science and Technology. Specifications of the MCC-3 instrument are as follows [2],

Sample heating rate: 0.1–10 K s^−1^Gas flow rate: 50 to 200 cm^3^ min^−1^, response time of <0.1 s, sensitivity of 0.1% of full scale.Repeatability is ±0.2% of full scale and an accuracy of ±1% of full-scale deflection.Sample size: 0.5–50 mg (milligrams).Detection limit: 5 mW.Repeatability: ±2% (10 mg specimen).

Pyrolyzer heating temperature was set from 75 to 700 °C, and combustor temperature was set at 900 °C for pyrolysis gas combustion. All tests were conducted following Method A and Method B, separately. Thirteen heating rates, 0.1, 0.2, 0.5, 1.0, 1.5, 2.0, 2.5, 3.0, 3.5, 4.0, 4.5, 5.0, and 5.5 K s^−1^, were performed for both standard methods.

In the Method A procedure the specimen undergoes an atmosphere-controlled thermal decomposition [2] when subjected to controlled heating in an oxygen-free/anaerobic environment. The gases released by the specimen during operation are swept from the specimen chamber by nitrogen, subsequently mixed with excess oxygen, and then completely oxidized in a high temperature combustion furnace. The volumetric flow rate and volumetric oxygen concentration of the gas stream exiting the combustion furnace are continuously measured during the test to calculate the rate of heat release by means of oxygen consumption. In Method A, maximum specific heat release rate *Q*_max_, which is maximum value of the specific heat release rate recorded during the test; heat release temperature *T*_max_, which is the specimen temperature at which the specific heat release rate is a maximum during controlled thermal decomposition; and specific heat release of sample *h*_c_ which is the net heat of complete combustion of the volatiles liberated during controlled thermal decomposition per unit initial specimen mass, were obtained at different heating rates.

In Method B procedure [18], the specimen is subjected to controlled heating in an oxidizing/aerobic environment. The specimen gases evolved during the controlled heating program are swept from the specimen chamber by the oxidizing purge gas and mixed with additional oxygen. The volumetric flow rate and volumetric oxygen concentration of the gas stream exiting the combustion furnace are continuously measured during the test to calculate the specific combustion rate by means of oxygen consumption. In Method B, maximum specific combustion rate *Q*_max_°, which is the maximum value of the specific combustion rate recorded during the test; combustion temperature *T*_max_°, which is the specimen temperature at which the specific combustion rate is a maximum during controlled thermal oxidative decomposition; and net calorific value of sample *h*_c_°, which is the net heat of complete combustion of the specimen measured during controlled thermal oxidative decomposition per unit initial specimen mass at different heating rates, were obtained.

### 2.2. Materials

Wood dust from Slovakian trees were harvested in central part of Slovakia. More details of these samples are listed in Table 1. The samples were taken from an automatic band saw used for construction and carpentry timber, prepared wood dust samples are shown in Figure 1. To obtain the required sample fraction size of 0.355–0.5 μm, the sieve analysis using the analytical sieve machine AS 200 Basic (ATS Scientific Inc., Burlington, NJ, USA) was applied. The samples were dried to 0% water/moisture content for further analysis using the Memmert UFB 500 Basic (Memmert GmbH+Co. KG, Schwabach, Germany). To determine the gross calorific value and heating value of the samples the bomb calorimeter IKAC 5000 control (Cole-Parmer Instrument Company Ltd., Vernon Hills, IL, USA) was used and STN ISO 1928:2009-07 [20] solid fuels applied, results are also listed in Table 1.

The gross caloric value *h*_0_ (kJ/kg) is defined as the heat released by perfect combustion (by the oxidation of the active elements of the fuel) of 1 kg of fuel and cooling of the flue gas and the ash to the initial temperature (i.e., 20 °C), while the water vapor condenses and changes to water. The gross caloric value is determined by experiments in a calorimeter [21]. The heating value hu (kJ/g) is defined as the heat released by the perfect combustion of 1 g of fuel, when the combustion is cooled to its original ambient temperature (20 °C), while the water (evaporated from the fuel produced by the oxidation of hydrogen contained in fuel and supplied with humid air) remains in the gaseous state. Before testing, the samples were dried at 103 ± 2 °C to reach the moisture content of 0% and further conditioned in a desiccator at the temperature of 20 ± 1 °C for 24 h. In the calculations of heating value, the relative moisture content of 8% was used.

## 3. Analysis of Test Results

### 3.1. Curve Shape Observations of Method A and Method B

Taking the BB sample as an example, the specific heat release heat curves with Methods A and B for 13 heating rates are drawn in Figure 2 and Figure 3. Apparently, there was only one single peak for Method A, while shoulder peaks could be found in Method B. Figure 4 shows the same heating rate 1 K s^−1^ case for both standard testing methods. Both curves had the same onset temperature while showing different shapes. The second peak for Method B was caused by the oxidation of carbon with the generation of carbon monoxide, which was generated during the pyrolysis procedure in the pyrolyzer room. However for Method A, there was no oxygen in the pyrolysis room, and accordingly there was no oxidation reaction, so the curve only showed one peak.

### 3.2. Typical Parameters Directly from Method A and Method B

After the MCC experiments, there would be still some residual existing in the crucible for the Method A test, while there was no residual after Method B. Oxygen was involved in the pyrolysis process of Method B, and the main component of the pyrolysis residue was a carbon residual, which can directly react with oxygen to produce carbon dioxide or carbon monoxide. However, in the process of Method A, there was no oxygen involved in the pyrolysis process, so some residual would still remain in the crucible. Figure 5 shows the average residues of the different samples, and it was found that the averaged residue for Method A was around 13.8%.

Table 2 shows the peak heat release rates and the corresponding temperature data of the materials with different heating rates by the measurements of Method A and Method B. The peak heat release rate can be regarded as the maximum point at which heat is released. It was found that the peak heat release rate and its corresponding temperature for Method A were always higher than those for Method B. From Table 2, we found that MCC combustion experiments with heating rates 5.5 K s^−1^ were close to the real burning face heating rates during polymer burning. Actually, the peak heating releases of 4.0, 4.5, 5.0, 5.5 K s^−1^ were the heat release upper limits for these wood MCC experiments, which were reflected in the oxygen content during the experiment, the oxygen concentration was still more than the 10% specified by the MCC experimental procedure.

### 3.3. Derived Parameters from MCC Tests

#### 3.3.1. Total Heat Release

Table 3 and Figure 6 show the specific heat release of sample *h*_c_ and the net calorific values of all 13 samples with Method A and B. It was apparent that the total heat release of Method A for each sample was smaller than that of Method B. Such a difference between the two methods is caused by their different pyrolysis procedures. In the procedure of Method A, since there is no oxygen existing during the pyrolysis process with the temperature heating from 75 to 700 °C, in this range some carbon residual will be generated. These carbon residuals in Method A find it hard to have any further oxidizing reactions.

However, for the procedure of Method B, there is oxygen existing during the pyrolysis process within 75~700 °C. The carbon residual reacts easily with oxygen to form carbon monoxide or carbon dioxide, which is the reason that there was no residual in the crucible after the test. Since the carbon residual takes the oxidizing reaction in Method B, the total heat release is larger than that of Method A, as shown in Table 3. Here, the Grubbs’ criterion was adopted to analyze the data set of THR and exclude gross errors. Low heating rate tests lead to large errors due to the low signal to noise ratio, so we analyzed the data with heating rates larger than 1 K s^−1^. The averaged difference of Method B larger than A was 4.77 kJ g^−1^. At the same time, it should be noted that the value of total heat release rate obtained by Method B was closer to the value of the combustor than that obtained by Method A, so Method B is more likely to be used as an alternative method to establish a relationship with the combustor.

Then, in the following part we tried to calculate the total heat release difference between the two methods, based on the theory that such a difference is caused by the oxidizing reaction of the carbon residual. The initial mass was 10 mg, and by calculation the total carbon element in wood (expressed as (C_6_H_10_O_5_)_n_) sample was 1.388 mg. These carbon residuals caused the total heat release difference between the two methods. Assuming that all this carbon can then be oxidized in Method B, then the total heat release difference can be expressed as $\nabla h_{c}=\frac{m_{c}Q_{CO2}}{n_{c}m_{0}}=4.0{\;\mathrm{kJ}\;g}^{-1}$, by theoretical calculation which is very close to the averaged experimental difference 4.77 kJ g^−1^.

#### 3.3.2. Ignition Capacity

The ignition capacity defines the ability of a material to resist ignition when it is exposed to an ignition source. The ignition potential of a material is measured by analyzing the heat released by combustion and the temperature required to initiate ignition in a given material. The ignition temperature is usually obtained by estimating the temperature corresponding to the release of 20 W/g of heat in the burning process. Alternatively, the temperature that corresponds to the 5% value of the converted integral of the heat release rate against time curves can be used. The ignition capacity of a material is essential in assessing its resistance to fire and flame.

Heat release capacity is an intrinsic property parameter to describe the heat release capacity of material combustion, which can be used to evaluate the flammability or thermal hazard parameter of different materials. However, from the definition of HRC in the literature [21,22,23], its formula results for HRC calculation are heavily dependent on the heating rate of MCC, which is not an intrinsic property of combustible materials. Therefore, the 20 W/g and 5% total released heat arrival definitions have been introduced. In this paper, we recalculated the ignition capacity of the results from Method A and B. It was found that by these two new definition methods (20 W/g and 5%), the ignition capacity was no longer dependent on heating rates, as shown in Figure 7 and Table 4 and Table 5. However, from Table 4 and Table 5 we found that the IGCs are only independent for heating rates larger than 1 K s^−1^, while for other smaller heating rates, this conclusion is not applicable, due to poor robustness.

#### 3.3.3. Heat Release Capacity

MCC uniquely measures the heat release capacity (HRC), a parameter that relates material properties to fire experiments. HRC characterizes the propensity of a material to release heat when subjected to thermal energy. The parameter measures the maximum heat release potential of a burning material in units of J g^−1^ K^−1^ and has proven to be an excellent predictor of fire response over the years. The HRC value of a material is estimated from the MCC measurements by calculating the maximum specific heat release rate per rate of temperature rise during the experiment. An elemental analysis by Keshavarz [23] showed that HRC could be obtained from the structural elements of the materials, such as the quantity of carbon, hydrogen, nitrogen, oxygen, chlorine and silicon atoms embedded in the material. For every material, the lower the HRC the better the flammability or fire resistance. HRC is one of the most extensively used parameters in assessing the heat release properties of polymeric materials. The heat release capacity ηc is a flammability parameter measured in Method A that is unique to this test method [19]. It is independent of the form, mass, and heating rate of the specimen as long as specimen temperature is uniform at all times during the test [20]. It is calculated by the maximum specific heat release rate during a controlled thermal decomposition divided by the heating rate in the test. However, this algorithm has been proved to be strongly dependent on the heating rate [24,25,26]. A new algorithm, as shown in Figure 8 and Table 6, was proposed in order to obtain robust results.

The limiting oxygen index (LOI) is the minimum concentration of oxygen which will support the combustion of a polymer. The LOI values of wood are generally between 23–26. Researchers proposed empirical formulas between LOI and HRC, LOI = 12% + 4000/HRC, LOI = 125/HRC [7,9]. However, the LOI value and HRC value do not fit such relations. Therefore, the relationship between HRC and LOI using existing materials cannot be described, and a large number of experiments need to be further analyzed in future.

## 4. Conclusions

The basic parameters, such as the combustion heat of wood dust, are helpful to support fire safety prevention and energy utilization. The results of this paper have a good evaluation on the MCC calorimetric evaluation method, and have a high guiding significance in wood flammability. Based on the MCC study of a variety of woods, the following conclusions are summarized:The experimental results obtained by Methods A and B are different, which are mainly manifested in the curves of the heat release rate curve, total heat released, peak temperatures, peak heat release rates, etc. The parameter peak values of Method A are higher than those of Method B, and the corresponding temperatures are also higher than those of Method B.The total released heat of Method B is close to that obtained by the oxygen boom combustion method. So, Method B has more possibility of replacing the oxygen boom method compared with Method A. The predicted value of the combustion heat obtained from Method B is more reliable since the total heat released by Method B is more referable.MCC combustion experiments with heating rates of 5.5 K s^−1^ are close to real burning face heating rates during polymer burning. Peak heat release at 4.0, 4.5, 5.0, 5.5 K s^−1^ reaches the heat release upper limits for these wood MCC experiments, which can be reflected in the oxygen content during the experiment, the oxygen concentration is still more than the 10% specified by MCC experimental procedure.

## Figures and Tables

**Figure 1 polymers-14-00045-f001:**
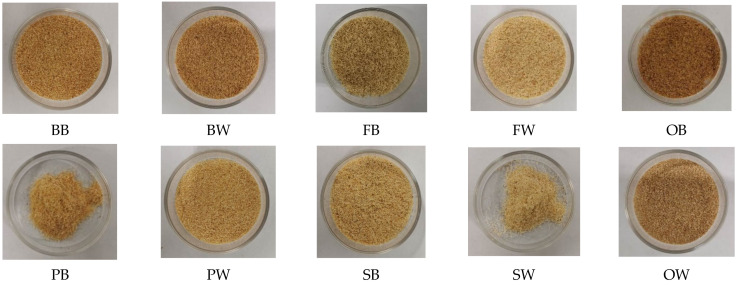
Slovakian wood dust of ten samples employed in this study.

**Figure 2 polymers-14-00045-f002:**
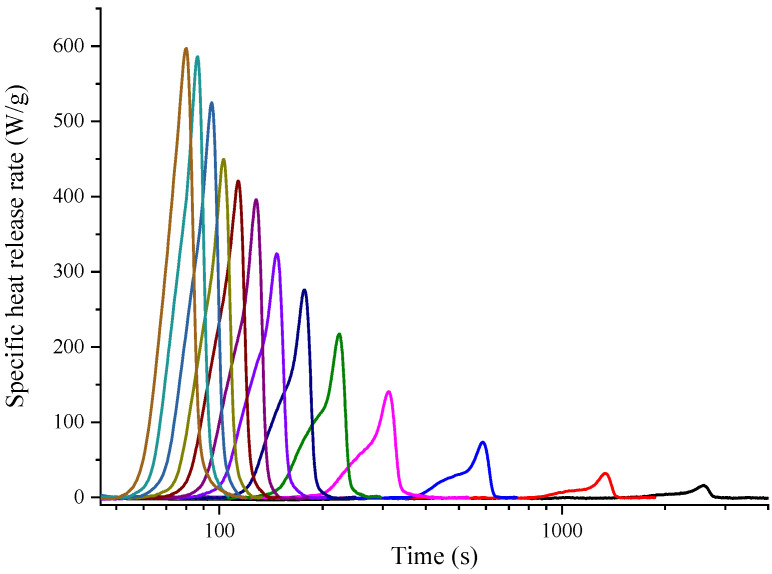
Specific heat release rate curve of BB sample with 13 heating rates ranging from 0.1 to 5.5 K s^−1^ by MCC Method A. (Different color means different heating rates. From right to left is 0.1, 0.2, 0.5, 1.0, 1.5, 2.0, 2.5, 3.0, 3.5, 4.0, 4.5, 5.0, and 5.5 K s^−1^.).

**Figure 3 polymers-14-00045-f003:**
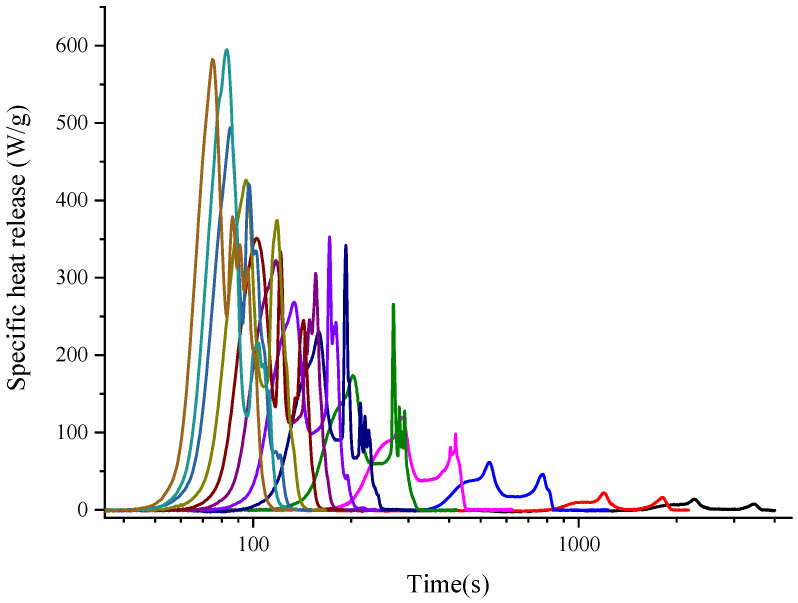
Specific heat release rate curve of BB sample with 13 heating rates ranging from 0.1 to 5.5 K s^−1^ by MCC Method B. (Different color means different heating rates. From right to left is 0.1, 0.2, 0.5, 1.0, 1.5, 2.0, 2.5, 3.0, 3.5, 4.0, 4.5, 5.0, and 5.5 K s^−1^.).

**Figure 4 polymers-14-00045-f004:**
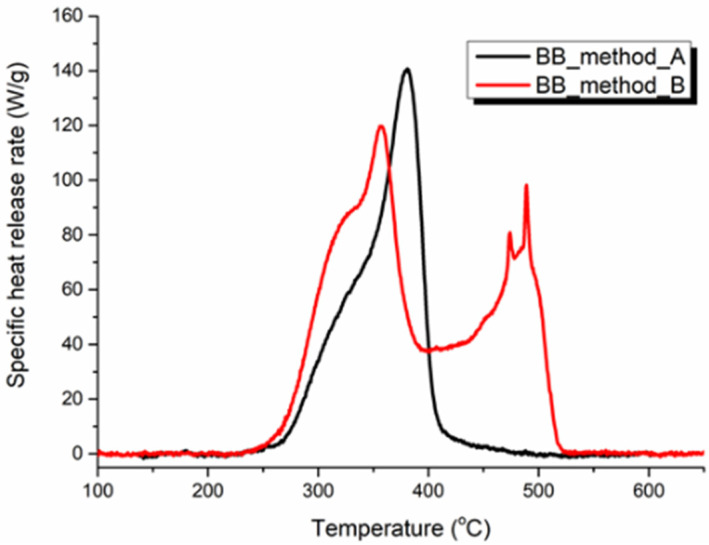
Comparison of specific heat release rate curve of BB sample with 1 K s^−1^ heating rate for Method A (black solid line) and Method B (red solid line).

**Figure 5 polymers-14-00045-f005:**
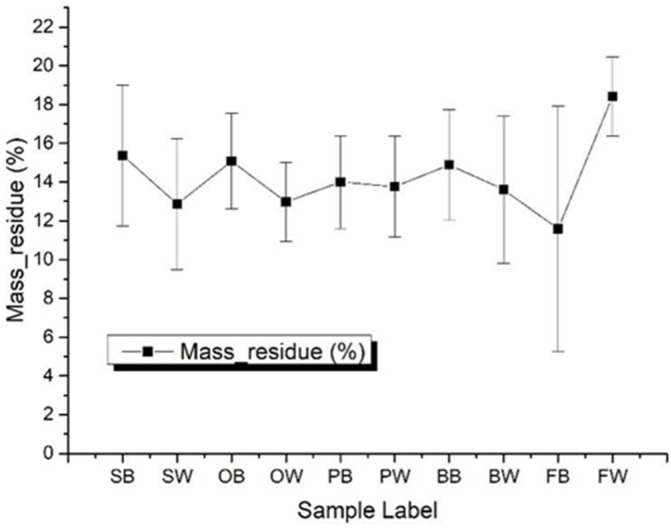
Residual mass percentage of each sample averaged by all heating rates with Method A.

**Figure 6 polymers-14-00045-f006:**
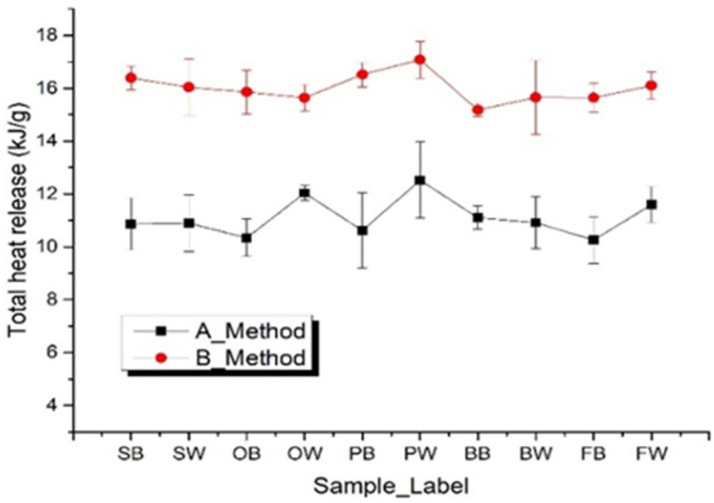
Total heat release of all thirteen samples with comparison of Methods A and B.

**Figure 7 polymers-14-00045-f007:**
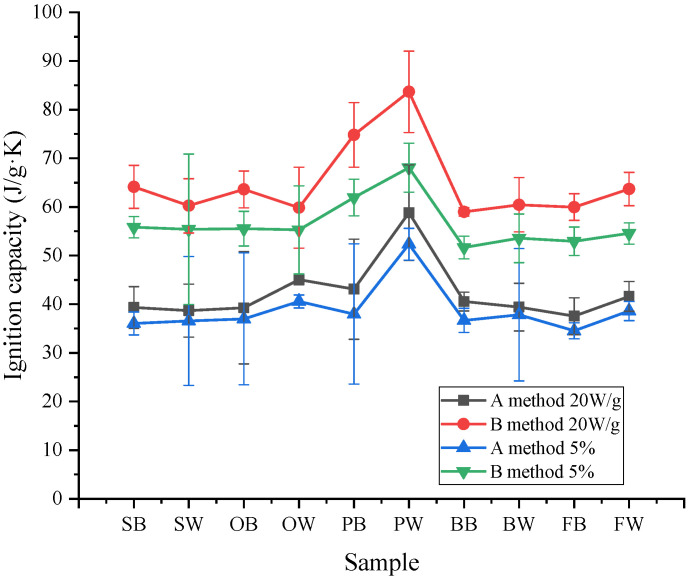
Ignition capacity defined by 20 W/g and 5% total heat release for results by Method A and B.

**Figure 8 polymers-14-00045-f008:**
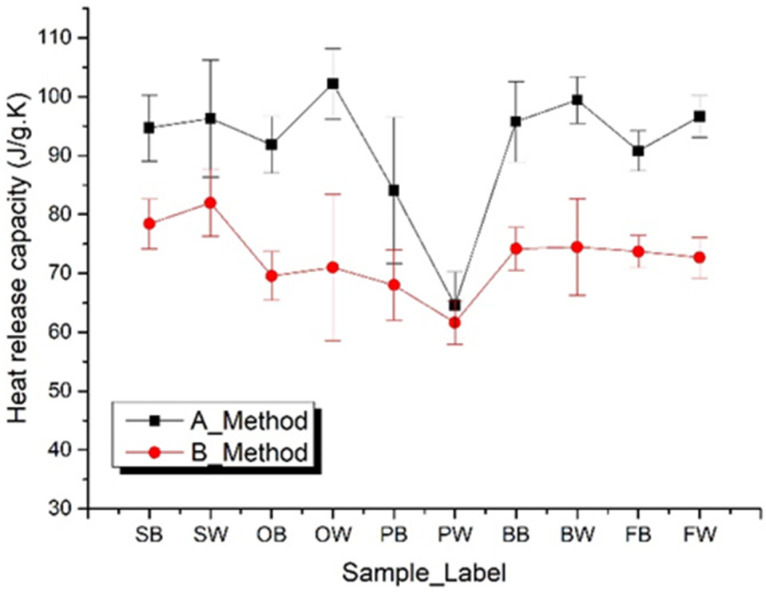
Heat release capacity defined by 20 W/g and 5% total heat release for results by Methods A and B.

**Table 1 polymers-14-00045-t001:** Type of wood dust and results from bomb calorimetric.

Sample Type	Label	H_0_(8)	Se	H_0_ (0)	Se	H_u_(8)	Se	H_u_(0)	Se
Spruce with bark	SB	17.8	0.31	19.3	0.34	16.3	0.31	18.0	0.34
Spruce without bark	SW	17.4	0.38	18.7	0.41	15.8	0.38	17.4	0.41
Oak with bark	OB	17.0	0.18	18.4	0.20	15.6	0.18	17.1	0.19
Oak without bark	OW	17.7	0.11	19.2	0.12	16.3	0.11	17.9	0.12
Pine with bark	PB	17.7	0.14	19.2	0.18	16.4	0.20	17.9	0.30
Pine without bark	PW	17.5	0.05	19.0	0.10	16.1	0.12	17.6	0.20
Beech with bark	BB	16.2	0.19	17.6	0.17	14.8	0.19	15.8	0.68
Beech without bark	BW	16.0	0.08	17.5	0.07	14.6	0.08	16.2	0.09
Fir with bark	FB	18.4	0.10	19.9	0.10	16.9	0.10	18.6	0.12
Fir without bark	FW	18.1	0.10	19.7	0.09	16.7	0.10	18.3	0.13

H_0_ (8) calorific value at self-defined sample moisture, i.e., 8% in this case, kJ g^−1^. H_0_ (0) calorific value at 0% sample moisture, kJ g^−1^. H_u_ (8) heating value at self-defined sample moisture, i.e., 8% in this case, kJ g^−1^. H_u_ (0) heating value at 0% sample moisture, kJ g^−1^. Se: standard error.

**Table 2 polymers-14-00045-t002:** Maximum specific heat release rate *Q*_max_ and maximum specific combustion rate *Q*_max_° (W/g) with corresponding temperature *T*_max_, and *T*_max_° (°C).

	Method	*β* (K/s)	0.1	0.2	0.5	1.0	1.5	2.0	2.5	3.0	3.5	4.0	4.5	5.0	5.5
BB	A	*Q* _max_	16.3	32.0	73.4	136.4	217.5	276.0	324.0	395.9	420.9	449.6	524.7	585.9	597.0
		*T* _max_	340.4	350.2	368.5	381.5	392.9	399.4	404.0	410.8	412.6	419.7	423.6	426.4	422.7
	B	*Q*_max_°	14.1	22.2	61.6	120.0	173.8	231.3	269.5	322.4	353.8	427.4	494.2	598.4	583.7
		*T*_max_°	307.8	319.9	341.3	357.3	362.5	367.9	374.4	385.2	377.3	389.3	383.9	403.9	399.5
BW	A	*Q* _max_	18.6	32.0	71.9	137.9	212.3	265.3	325.5	397.7	447.9	472.7	567.3	620.0	699.2
		*T* _max_	337.8	349.6	369.4	383.0	391.2	398.6	404.1	407.9	413.8	415.0	421.7	420.7	427.2
	B	*Q*_max_°	13.9	26.8	57.1	109.8	191.0	197.6	271.3	321.4	357.1	438.2	527.0	519.2	522.4
		*T*_max_°	308.2	322.5	340.8	355.8	366.2	375.8	379.4	386.8	391.8	388.2	393.0	402.4	399.5
SB	A	*Q* _max_	14.1	23.1	69.4	134.0	192.7	248.1	293.5	357.6	380.4	479.0	547.9	611.8	664.9
		*T* _max_	341.5	354.4	374.3	388.8	398.7	404.4	409.7	417.1	422.4	417.6	427.9	429.0	432.4
	B	*Q*_max_°	16.3	29.1	83.1	163.1	237.5	294.0	351.2	413.6	478.9	543.5	636.2	688.7	693.5
		*T*_max_°	312.9	326.1	347.2	361.9	370.1	379.7	387.8	390.1	394.0	399.0	403.8	398.5	408.5
SW	A	*Q* _max_	16.4	29.9	73.1	142.3	188.9	264.0	327.7	322.0	460.3	510.9	497.6	569.0	644.9
		*T* _max_	350.0	363.5	383.5	397.8	406.9	412.6	419.3	426.4	430.4	434.3	438.9	440.0	439.4
	B	*Q*_max_°	27.2	55.9	107.9	209.4	301.5	412.1	430.8	531.4	588.1	628.1	757.2	828.7	819.4
		*T*_max_°	314.5	327.3	351.0	366.5	382.0	388.1	399.4	398.6	407.2	408.9	414.3	408.2	423.2
OB	A	*Q* _max_	18.4	25.5	55.0	123.0	170.2	234.7	290.4	336.0	397.9	406.9	467.9	508.3	524.6
		*T* _max_	327.3	342.9	354.1	369.3	374.7	382.9	388.3	393.4	390.7	399.2	402.3	404.9	409.4
	B	*Q*_max_°	38.1	30.3	70.7	136.5	220.5	291.5	350.3	384.5	447.0	492.2	558.6	624.3	676.2
		*T*_max_°	311.1	317.3	335.7	348.0	363.6	364.4	376.8	380.3	381.0	379.3	383.7	389.9	391.5
OW	A	*Q* _max_	12.5	30.4	71.3	148.8	201.8	262.4	328.2	397.8	415.4	465.3	545.3	586.0	658.6
		*T* _max_	346.9	357.1	373.0	390.0	396.5	404.6	408.5	414.6	416.5	420.7	423.4	427.6	429.4
	B	*Q*_max_°	21.7	48.5	94.0	180.2	250.8	316.7	395.3	420.1	484.1	524.9	600.5	687.0	666.9
		*T*_max_°	310.2	322.9	345.4	357.3	373.2	379.9	383.9	396.8	401.8	403.0	406.8	409.5	408.3
PB	A	*Q* _max_	14.7	25.9	67.5	121.1	196.0	252.0	303.0	354.3	404.2	461.8	484.3	535.5	576.3
		*T* _max_	351.4	363.9	382.3	395.7	407.8	410.8	418.3	422.0	428.3	433.3	437.5	439.7	444.0
	B	*Q*_max_°	22.2	40.2	85.6	189.1	273.8	342.1	397.1	485.6	536.5	599.0	606.8	629.2	709.3
		*T*_max_°	312.8	326.7	349.9	366.2	376.7	385.4	388.5	394.0	399.2	402.9	407.4	407.2	418.6
PW	A	*Q* _max_	21.9	26.3	64.5	123.2	199.4	259.6	302.0	359.0	378.9	486.3	523.4	562.1	616.7
		*T* _max_	347.8	363.1	383.5	397.8	407.6	416.0	423.0	427.2	430.7	436.5	437.8	441.1	443.8
	B	*Q*_max_°	17.3	34.5	79.6	173.2	235.3	343.4	409.7	431.6	517.1	554.3	662.9	636.9	674.8
		*T*_max_°	315.0	329.0	351.4	370.0	381.7	392.5	395.9	403.8	406.4	408.9	411.8	420.4	426.5
FB	A	*Q* _max_	14.3	26.2	63.6	122.8	186.9	265.5	323.8	383.0	452.1	484.4	562.1	567.7	661.3
		*T* _max_	336.8	352.2	367.9	382.8	389.4	398.1	405.8	410.2	411.3	418.5	418.5	416.9	426.4
	B	*Q*_max_°	14.1	30.5	70.2	133.6	190.9	253.7	305.7	356.7	403.4	476.4	464.1	561.9	672.2
		*T*_max_°	310.9	324.3	342.6	356.6	365.4	374.9	378.4	386.4	388.9	386.4	402.0	398.6	401.3
FW	A	*Q* _max_	15.2	25.0	61.7	128.5	175.0	241.4	292.4	385.2	436.8	471.8	526.4	565.4	614.8
		*T* _max_	329.8	345.5	363.6	378.3	385.2	391.1	396.5	408.5	408.9	415.8	420.3	415.1	429.0
	B	*Q*_max_°	11.4	19.9	51.4	1174.8	162.8	219.7	223.1	317.1	381.5	421.2	492.2	526.3	516.0
		*T*_max_°	312.2	323.8	342.6	356.1	358.1	360.6	367.1	388.9	382.0	393.3	390.6	394.2	395.9

**Table 3 polymers-14-00045-t003:** Specific heat release of sample *h*_c_ and net calorific value of sample *h*_c_° (kJ/g).

	*β* (K/s)		0.1	0.2	0.5	1.0	1.5	2.0	2.5	3.0	3.5	4.0	4.5	5.0	5.5
SB	A	*h* _c_	-	-	9.3	10.4	10.2	10.8	10.3	10.5	10.0	11.7	12.1	11.9	12.3
	B	*h*_c_°	-	-	-	16.3	16.3	15.6	16.5	16.5	16.8	15.8	16.7	16.3	17.1
SW	A	*h* _c_	11.0	8.6	10.1	10.8	10.1	10.9	11.6	9.7	12.0	12.3	11.0	11.3	12.3
	B	*h*_c_°	13.8	16.7	13.7	15.9	16.5	16.9	15.9	16.5	16.4	16.2	16.6	16.5	16.9
OB	A	*h* _c_	-	9.2	9.0	10.7	9.9	10.8	10.3	10.9	11.5	10.2	10.7	10.5	10.4
	B	*h*_c_°	-	-	-	14.5	16.2	17.6	15.9	15.1	15.3	16.1	16.0	15.8	16.1
OW	A	*h* _c_	-	-	-	11.8	11.6	12.0	12.1	12.2	11.6	12.2	12.4	12.4	12.1
	B	*h*_c_°	-	-	-	15.4	15.5	15.5	16.0	15.2	15.7	15.0	15.3	16.6	16.2
PB	A	*h* _c_	8.4	8.1	9.2	9.5	9.9	11.9	10.7	11.8	11.9	11.9	11.7	11.2	11.8
	B	*h*_c_°	-	16.0	-	15.9	17.2	16.6	16.3	16.5	16.4	17.3	16.8	16.0	16.7
PW	A	*h* _c_	-		10.0	10.2	11.8	13.0	13.0	12.3	12.3	12.9	13.8	14.0	14.5
	B	*h*_c_°	-	-	-	16.8	16.5	17.8	18.3	16.8	16.2	17.2	17.9	16.5	16.8
BB	A	*h* _c_	-	-	-	10.3	11.5	11.3	11.5	11.3	11.1	10.3	11.3	11.2	11.2
	B	*h*_c_°	-	-	-	14.9	15.3	15.3	15.3	15.6	15.1	15.5	14.9	14.9	15.0
BW	A	*h* _c_	10.9	9.0	9.1	10.6	10.9	11.0	11.1	11.9	11.3	10.3	11.8	11.9	12.1
	B	*h*_c_°	-	-	12.4	15.4	18.5	15.7	16.1	16.0	15.7	15.5	15.8	15.6	15.4
FB	A	*h* _c_	-	-	8.6	8.8	9.5	10.5	10.5	10.5	11.0	10.6	10.9	10.6	11.3
	B	*h*_c_°	-	-	14.7	15.3	15.6	16.2	16.4	16.4	15.6	15.6	15.1	15.4	15.8
FW	A	*h* _c_	-	-	-	10.5	10.3	11.7	11.6	12.4	11.8	11.6	11.6	12.4	11.6
	B	*h*_c_°	-	-	-	15.5	15.9	16.2	15.1	15.8	16.4	16.8	16.5	16.4	16.4

**Table 4 polymers-14-00045-t004:** Ignition capacity calculation for ten wood samples with Method A and Method B by 20 W/g definition.

*β* (K/s)		0.1	0.2	0.5	1.0	1.5	2.0	2.5	3.0	3.5	4.0	4.5	5.0	5.5
SB	A	-	-	31.8	37.2	36.4	39.6	37.6	37.5	35.9	43.5	43.8	43.6	45.6
	B	-	-	-	59.3	60.5	58.4	64.5	63.8	67.2	61.4	69.2	64.9	72.0
SW	A	-	26.3	34.2	38.1	35.6	39.3	42.3	34.3	43.6	45.0	39.5	41.0	44.9
	B	48.0	58.7	48.8	60.3	62.7	64.7	59.6	63.5	62.0	62.0	63.1	64.9	64.7
OB	A	-	29.9	34.5	41.4	38.2	42.4	39.9	41.9	45.3	38.6	40.6	40.0	38.4
	B	-	-	-	58.4	65.2	71.7	63.4	59.0	60.9	65.2	64.1	63.1	65.0
OW	A	-	-	-	44.0	43.7	45.7	45.0	45.6	43.4	46.1	46.1	46.4	44.5
	B	-	-	-	59.4	61.3	59.2	62.9	60.0	59.9	59.6	45.2	78.4	52.6
PB	A	-	30.9	24.5	33.0	34.4	54.2	39.1	53.8	52.3	47.9	52.8	45.4	48.7
	B	-	55.4	-	61.2	81.8	77.1	71.3	77.7	75.5	85.3	79.4	77.3	81.0
PW	A	-	-	33.0	51.5	58.1	64.8	64.9	60.6	54.8	62.0	65.4	65.5	66.4
	B	-	-	-	64.1	80.9	88.2	92.7	84.9	77.8	87.6	92.6	82.5	85.3
BB	A	-	-	-	38.3	43.0	41.7	42.5	41.0	40.5	36.5	41.2	39.8	40.9
	B	-	-	-	57.8	59.2	59.9	58.8	59.7	58.4	60.3	59.7	58.1	58.0
BW	A	-	28.6	32.6	36.5	40.7	40.7	41.1	44.7	41.3	36.6	42.8	43.9	43.4
	B	-	-	47.8	60.1	72.4	60.5	62.2	62.0	60.3	60.1	61.2	59.2	59.1
FB	A	-	-	30.0	31.4	34.9	38.9	38.3	38.7	40.4	39.1	40.5	39.2	41.5
	B	-	-	54.6	58.7	60.2	62.3	63.7	63.2	60.1	61.0	56.7	58.6	60.6
FW	A	-	-	-	37.3	35.9	43.1	42.8	44.5	42.4	41.4	41.2	46.1	41.4
	B	-	-	-	60.2	62.8	64.9	57.6	60.3	64.2	65.8	67.0	68.7	65.1

**Table 5 polymers-14-00045-t005:** Ignition capacity calculation for ten wood samples with Method A and Method B by 5% heat release definition.

*β* (K/s)		0.1	0.2	0.5	1.0	1.5	2.0	2.5	3.0	3.5	4.0	4.5	5.0	5.5
SB	A	-	-	33.3	37.0	34.6	36.7	34.5	33.9	32.4	38.4	38.5	38.0	39.1
	B	-	-	-	59.6	56.2	53.6	56.9	56.5	56.8	52.1	56.3	53.4	57.0
SW	A	43.2	30.8	36.2	37.5	34.3	36.5	38.4	31.3	38.6	39.4	34.7	35.7	38.7
	B	54.9	64.7	49.8	56.4	56.6	65.3	52.5	54.8	53.1	52.6	52.9	53.0	53.6
OB	A	-	37.6	35.9	40.3	36.5	39.2	36.6	37.9	39.9	34.7	36.0	35.3	34.0
	B	-	-	-	54.9	59.4	63.7	55.9	51.7	52.8	55.7	54.2	53.1	53.8
OW	A	-	-	-	42.7	41.1	42.2	41.0	41.0	38.8	40.2	40.2	40.1	38.4
	B	-	-	-	49.7	52.0	51.3	55.4	54.4	55.8	57.3	44.5	78.9	53.5
PB	A	31.6	30.5	32.7	33.0	32.6	48.5	36.2	45.9	43.5	40.5	42.4	37.2	39.2
	B	-	61.4		60.8	70.6	64.0	59.2	62.0	58.8	65.6	60.1	59.4	59.5
PW	A	-	-	51.3	47.7	53.9	56.8	55.8	51.8	46.0	51.0	54.4	54.2	52.8
	B	-	-	-	72.9	69.6	72.5	74.8	68.3	59.5	66.9	70.4	62.0	63.7
BB	A	-	-	-	37.2	40.7	38.9	39.1	33.1	37.0	33.1	36.5	35.4	35.8
	B	-	-	-	54.6	54.5	53.7	52.6	52.5	50.8	51.3	50.1	48.7	47.8
BW	A	45.3	37.4	33.6	35.7	38.8	37.9	38.0	40.6	37.0	33.0	38.2	38.4	38.0
	B	-	-	47.6	56.9	66.0	54.9	55.5	54.2	52.0	50.6	51.8	50.3	49.3
FB	A	-	-	32.1	31.2	33.5	36.2	35.2	34.8	36.2	34.6	35.5	34.5	36.0
	B	-	-	54.9	55.4	55.3	55.8	56.0	54.6	51.4	51.5	48.1	49.1	50.1
FW	A	-	-	-	37.6	35.9	41.4	40.1	40.7	39.0	37.3	37.4	40.5	36.3
	B	-	-	-	56.9	57.0	57.6	50.9	52.3	54.6	55.0	54.4	54.2	52.6

**Table 6 polymers-14-00045-t006:** Heat release capacity calculation results for ten wood samples with Method A and Method B.

*β* (K/s)		0.1	0.2	0.5	1.0	1.5	2.0	2.5	3.0	3.5	4.0	4.5	5.0	5.5
SB	A	-	-	104.8	99.1	99.5	95.2	91.8	93.7	82.7	93.0	94.5	96.3	91.3
	B	-	-	-	80.3	85.7	79.5	78.6	82.0	78.3	74.1	73.7	80.1	71.7
SW	A	116.1	110.1	104.1	101.8	90.4	95.4	96.7	81.9	97.6	95.4	84.4	86.1	92.0
	B	79.5	93.4	72.3	87.8	83.3	75.3	82.7	83.8	81.1	77.1	81.7	88.1	79.4
OB	A	-	95.5	86.7	99.7	92.4	97.7	95.1	92.8	94.0	87.3	89.0	88.0	84.0
	B	-	-	-	67.5	69.7	78.2	68.2	64.5	69.5	73.7	71.0	64.2	69.1
OW	A	-	-	-	106.4	103.8	101.0	104.2	103.9	94.9	97.3	98.5	96.9	115.3
	B	-	-	-	64.1	66.5	66.2	71.7	70.5	70.3	73.0	55.7	103.5	68.5
PB	A	109.2	88.8	97.8	90.5	98.7	73.5	87.2	70.7	74.9	82.0	70.2	75.5	74.3
	B	-	81.9	-	70.4	67.3	68.6	70.9	68.2	67.2	69.8	62.9	59.8	61.1
PW	A	-	-	54.5	57.3	63.9	68.5	68.5	64.6	58.1	68.0	70.4	70.8	66.2
	B	-	-	-	61.9	60.4	66.1	67.4	58.9	63.4	61.3	64.0	56.2	56.7
BB	A	-	-	-	97.6	104.8	105.0	102.8	86.1	95.2	89.9	94.3	94.6	87.5
	B	-	-	-	71.5	76.5	77.3	78.9	77.3	70.1	76.2	74.2	68.2	71.5
BW	A	96.2	89.6	101.3	98.9	104.3	100.9	99.8	101.4	101.9	94.7	99.8	100.5	103.7
	B	-	-	62.4	74.9	95.1	72.2	74.7	72.9	71.9	79.6	76.3	71.5	67.6
FB	A	-	-	91.9	90.0	91.6	95.3	94.8	90.8	94.3	86.5	91.6	85.3	86.9
	B	-	-	72.2	75.4	75.1	74.5	79.0	72.5	75.1	73.5	68.5	74.1	70.9
FW	A	-	-	-	102.0	95.8	95.0	93.9	103.3	98.4	95.6	94.4	92.8	95.3
	B	-	-	-	74.5	72.3	74.8	68.8	76.1	78.2	74.4	71.3	68.1	68.6

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
