# Peer review of "Wood Dust Flammability Analysis by Microscale Combustion Calorimetry"

_polymers, 2021, doi:10.3390/polym14010045_

Round 1
Reviewer 1 Report
The article needs revisions, my comments are below;
- The article is too wordy and needs a strong correlation with proposed Method A with Method B.
- The authors should comment on the wood chip dust properties and their characterizations.
- Very little literature review, merely 18 references. Deep analysis is needed along with strong literature review.
- The conclusion part is rather unscientific, of course the results would be different from both methods used. It should be revised and concise to main findings only.
Author Response
Kindly find the attached file for response to reviewer.

Reviewer 2 Report
The manuscript by Xu, Q. et al. describes on "Wood Dust Flammability Analysis by Microscale Combustion Calorimetry." The microcombustion calorimetry was used to study kinetics of combustion for Slovakian wood samples at 13 different heating rates. Ignition capacity and heat release capacity were measured for all wood samples. The comparison of specific heat release and net calorific heat of combustion with gross calorific value and heating value by MCC was made with those by bomb calorimeter. This study has the practical implication for wood dust flammability. Therefore, I find it interesting and worthy of publication in polymers.
Author Response
We thank the reviewer for positive and constructive feedback on our manuscript.
Reviewer 3 Report
The manuscript deals with the investigation and analysis of wood dusty flammability by using Microscale Combustion Calorimetry. The manuscript can be accepted for publication in the Polymers Journal after minor revision. Please, see below my comments on your work:
In general, the title (lines 2-3), abstract (lines 10 to 22) and the keywords (line 23) correspond to the title, aims and objectives of the manuscript.
In lines 10-12, I’d recommend to revise the first sentence of the abstract, i.e. “To study the practicability of micro combustion calorimeter to calorimetry kinetics of wood, 10 kinds of Slovakian wood samples were studied by micro combustion calorimeter with 13 heating rates from 0.1 to 5.5 K/s.” in order to avoid unnecessary repletion “To study….were studied”.
In line 11, please consider revising “Slovakian wood samples” – the species included in the study are not native to Slovakia only. I’d suggest to replace “Slovakian” with “common hardwood and softwood species”.
In line 12, please provide the full term, i.e. “Microscale combustion calorimeter”, followed by the abbreviation MCC. In addition, please provide more specific results obtained for the flammability characteristics (ignition capacity and heat release capacity) of the tested wood samples.
In general, the Introduction part is well written and informative, and provides relevant information and references on the research topic. I’d recommend to extend the part dealing with wood dust and associated health hazards, including flammability and dust explosion potential.
In line 107, please provide additional information on the microscale combustion calorimeter used (company producer, city, country).
In the Materials section, please provide more information on the tree species used in the research, e.g. provide the botanical names of the species to properly distinguish them.
Line 150: “The samples were sampled” – please revise to avoid repetition.
Line 151: please replace “lumber” with “timber” (British English).
In addition, please revise “…an automatic band saw used for construction and carpentry lumber, as shown in Fig. 1..”, since in Figure 1 we can see the wood dust samples only.
The inclusion of additional references, as recommended above, will significantly improve this section of the manuscript.
In line 154, please provide relevant information on the analytical sieve machine AS 200. The same remark applies to the Memmert UFB 500 Basic and IKAC 5000 equipment used.
In line 157, please add the standard STN ISO 1928:2003-07 also in the References of your manuscript.
In line 187, please adjust the labels of the photos in Figure 1.
Overall, the Materials and Methods section is well written and detailed, but can be further elaborated based on the comments provided.
In line 190, please replace “Take” with “Taking” in order to be grammatically correct.
If possible, please provide better quality images for Figures 2-4. The same comment applies also to Figure 7.
In line 230, the font of Table 2 title is different from the main text, please revise.
In general, the results of the study are detailed, well-presented and informative. I’d recommend to add more discussion of the results with relevant research works in the field.
The Conclusion part (lines 321-346) is specific and reflects the main findings of the manuscript. Please discuss the potential for future studies in the field, and practical applications of the results obtained.
The References cited are appropriate to the topic of the manuscript. Some of the references are not formatted in accordance with the journal requirements, please check the Instructions for authors.
Best regards!
Author Response
We thank the reviewer for positive and constructive feedback on our manuscript. Kindly find the attached response to reviewer file.

Round 2
Reviewer 1 Report
No more comments to add.
This manuscript is a resubmission of an earlier submission. The following is a list of the peer review reports and author responses from that submission.